# OpenReview forum: "MiraGe: Editable 2D Images using Gaussian Splatting"
_ICML.cc/2025/Conference — ICML 2025 poster_

### Official Review · Reviewer_vmYq · 2025-03-10

**Overall Recommendation:** 2

**Summary:**

This paper presents approaches to editing 2D images represented by Gaussian Splatting. The authors propose to use 3D flat Gaussians optimized with mirrored cameras from two opposite sides to represent 2D images, with quality better than other prior works. With the GS-represented images, this paper demonstrates different ways to do image editing with the assistance of various geometry processing tools.

## update after rebuttal

I appreciate the efforts made by the authors during the rebuttal. The authors’ responses addressed some of my concerns. The proposed method to fit Gaussians for representing and editing 2D images is somewhat effective. Regarding the editing capability, editing the image by manipulating augmented Gaussians is interesting, but I don’t think that manipulating Gaussians can make image editing simpler or more efficient. The authors should clearly state such limitations or tradeoffs in the paper. I will keep my borderline rating ("weak reject") on this submission, however, I won't have objections if other reviewers agree to accept this work.

**Claims And Evidence:**

In the following 3 questions, I assume the method and evaluation in this paper refer to the proposed approach to representing 2D images with 3D Gaussians.

The claims about their new approach to better fitting 3D Gaussians to represent 2D images are somewhat supported by their evaluations. However, the authors fail to give a clear reason why amorphous-MiraGe gives better editing quality, and how it resolves the artifacts shown in other variants.

**Essential References Not Discussed:**

References are sufficient.I reviewed all the parts.

**Experimental Designs Or Analyses:**

The comparisons with DragGAN (Fig. 8 & 10) and PhysGen (Fig. 12) are not sound to me. Although the proposed method may give better visual quality from those figures, the manual efforts to achieve MiraGe’s results are way more than DragGAN and PhysGen. This comparison is like classical PhotoShop vs. Generative image editing, which does not mean anything.

**Methods And Evaluation Criteria:**

The evaluation makes sense.

**Other Comments Or Suggestions:**

This paper is fundamentally an image processing paper. There are a few machine learning contents inside (e.g., new ways to fit 3DGS for 2D images) but not too many. I am concerned whether ICML is a good fit for this work, as opposed to other computer vision/graphics or image processing conferences.

**Other Strengths And Weaknesses:**

Strengths:
* Utilizing two opposite cameras to fit 3D Gaussians to represent 2D images is an interesting idea, and evaluation also shows the effectiveness of this idea.
* I appreciate the authors’ manual efforts in making all these image edits/animations.

Weaknesses:
* Once an image is represented as Gaussians, editing those Gaussians is quite trivial, since Gaussians here are just discrete 3D points that can be freely moved. The real difficulty is how to easily edit the 2D image with 3D Gaussians. When I read the title of this paper, I was expecting to see some novel ways to ease the Gaussian editing for 2D images. However, the editing part still relies on existing tools, which usually require a lot of manual effort to bind the Gaussian points and to precisely define deformation or motions. Moreover, regular images represented by pixels can also do these 2D editings with more mature tools/pipelines. Finally, I acknowledge the proposed improvement from amorphous-MiraGe with artifacts after editing compared to other variants. However, this comparison is only shown in a small figure (Fig. 6), which makes it hard to extensively evaluate the proposed improvements.

**Questions For Authors:**

None

**Relation To Broader Scientific Literature:**

This work is somewhat related to implicit neural representation (just a bit “neural” and “implicit”) and 2D/3D geometry processing.

**Theoretical Claims:**

There’s no proof for theoretical claims.

---

> ### Author Rebuttal · Authors · 2025-03-29
>
> We appreciate the Reviewer’s feedback. We also thank Reviewer for appreciating the concept and work: “utilizing two opposite cameras to fit 3D Gaussians to represent 2D images is an interesting idea, and evaluation also shows the effectiveness of this idea”.  Since the section for questions directed to us is empty, we have opted to refer to other sketches.  We have made every effort to resolve any ambiguities. We hope the additional clarifications provide further clarity. We remain open and keen to provide further explanations if needed.
>
> EDOA: The comparisons with DragGAN and PhysGen are not sound to me...
>
> We would like to mention that representing a 2D image using Gaussian splatting for editing is a new concept, and it is difficult to compare it with existing tools. Therefore, we apply comparisons with existing solutions such as DragGAN and PhysGen, which shows the potential of our approach.
>
> In the case of PhysGen, we chose these models to highlight their differences. An interesting case is PhysGen's input control, which offers limited editing capabilities. In PhysGen, the authors utilized a physics engine for training, with a diffusion model serving as an intermediary. However, this approach restricts user control. In contrast, we demonstrate that MiraGe allows direct use of a physics engine for Gaussian representation, eliminating the need for a complex diffusion model conditioned on the physics engine. Additionally, our representation is compatible with multiple engines and tools, including Taichi Elements and Blender.
> In summary our representation enables editing directly in 3D, making a comparison with simple pixel modifications insufficient. Ultimately, we use this comparison to highlight the strengths of our model and its potential applications.
>
> C&E/ W
>
> In 2D-MiraGe primitives are constrained to a 2D plane, meaning edits should only be made on a plane parallel to the selected one. Otherwise, during rotation, they will fade away, - much like a thin piece of paper viewed from the edge in real life. This is why 2D-MiraGe was designed for compatibility with 2D engines, similar to Graphine-MiraGe, which excels in layered editing scenarios.
>
> Amorphous-MiraGe, on the other hand, is not strictly confined to a single plane, giving it a natural 3D effect. From a practical standpoint, this makes it the most intuitive to edit in 3D space. If only 2D edits, such as simple rotations in one plane, were applied, there would be no noticeable difference between the model versions. This distinction is illustrated in Fig. 6. However, we acknowledge that the image is too small, and we will enlarge it to improve clarity. We thank Reviewer for pointing this out.
>
> We hope our explanation is sufficient, and we remain open to more questions.
>
> W:  (...) editing those Gaussians is quite trivial (..)
>
> As 3D Gaussian editing still advances, so do methods for reducing artifacts. A recent study (arXiv 03.2025) [1b] enforces spherical shapes, while PhysGaussian [2b] uses an Anisotropy Regularizer—both preventing sharp artifacts during rotations.
> In 2D achieving high-quality and robust nonlinear modifications images is still challenging, especially when performing affine transformations on low-resolution images, where inaccuracies can lead to the appearance of holes. However, with Gaussians, affine transformations preserve key properties such as color and opacity, allowing us to avoid numerical errors and eliminate this issue.  Moreover, the additional 3rd dimension can simplify animation creation in certain cases. For example, it facilitates the natural motion of a waving flag as illustrated in Fig. 3. We hope that the use of Gaussians for 2D image representation will continue to evolve, leading to more advanced and mature editing tools in the future. Our work demonstrates that such editing is not only possible but also promising.
>
> OC&S: (...) I am concerned whether ICML is a good fit for this work...
>
> In our paper, we introduce a novel approach to 2D image representation, specifically designed for manual editing and integration with physical engines. As the reviewers pointed out, our method lacks direct competitors - a fact that, in our view, underscores its originality.
>
> In the reconstruction task, our method achieves superior results in a shorter time. As discussed in our response to Reviewer u88z, W2, our approach outperforms the baselines in PSNR with just 5k iterations  Moreover, MiraGe enables a range of capabilities:
> - Integrating physical engines with 2D images
> - Editing 2D images directly within a 3D space
> - Enabling complex nonlinear modifications of 2D images
>
> Therefore, we believe ICML is a suitable venue for this work, as we present a new approach to representing 2D images.
>
> We are committed to improving our paper and greatly appreciate the feedback regarding the visibility of Fig. 6. We will address this concern and make the necessary adjustments to the camera-ready version.

---

> > ### Comment · Reviewer_vmYq · 2025-04-04
> >
> > Thanks for the rebuttal. I acknowledge the novelty of the method fitting Gaussians to represent 2D images. Metrics in the paper and the rebuttal are good to me. However, regarding the authors' responses on Gaussian editing, I have a few more comments:
> >
> > * EDOA: My point is that the proposed method cannot easily complete the editing task compared to generative models. For example, you only need to click a few keypoints to achieve the editing effect with DragGAN, but this method requires much more time and effort in meticulously manipulating lots of Gaussians to achieve similar editing effects.
> >
> > * C&E/ W: The authors in the rebuttal mentioned "intuitive to edit in 3D space". The proposed method only does the faithful reconstruction of 2D images. I don't see any clear evidence that MiraGe can preserve decent 3D information in the reconstructed image. Without good 3D structure/geometry/semantics, directly editing in the 3D space is not an easy job. Providing renderings of Gaussians from different side views of the image plane would be beneficial to support the claim.

---

> > > ### Author Response · Authors · 2025-04-05
> > >
> > > Thank you for your response. We are happy to address any concerns or provide further clarification.
> > >
> > > EDOA:
> > >
> > > We agree that generative models have some advantages in editing but also have limitations. When we apply deep generative models for editing, such models can hallucinate – changing small elements of the objects. This is shown in our paper in Fig. 10. The second row demonstrates that when generative models are used to modify the position of the legs, unintended alterations occur in the woman's face. We interpret this as a limitation of the generative approach.  If we want to force the model to produce the same object with modifications, it is not trivial and requires much time.
> > >
> > > Our MiRaGe model requires manual triangle modification (or the application of a physical engine), but it guarantees consistent changes.
> > >
> > > All models have advantages and limitations. Gaussian representation can also find applications in deep generative models, which can, in the future, integrate the strengths of both technologies.
> > >
> > > C&E
> > >
> > > Thank you for bringing this to our attention. This feedback allows us to refine our explanations for greater clarity. We agree that it is not trivial to represent a 2D image in 3D space. Consequently, we do not claim to produce a full 3D object. As noted in the introduction, our approach is grounded in human perception of 2D objects within a 3D environment, which we illustrate using the analogy of a sheet of paper (lines 047–048). We refer to 2.5D [1] for such modifications, which is well described in the literature and pertains to modifying 2D images in 3D space.
> > >
> > > In the appendix (lines 559–565), we clarify that "Editing images in pixel representation is a well-established technique with many existing solutions, and 2D image editing tools (e.g. Photoshop, GIMP). MiraGe presents a concept that combines 2D and 3D representations to achieve the 2.5D effect commonly used in video games and VR [1].”
> > >
> > > By ease of editing in 3D, we refer to the ability to manipulate such an image or photograph within a 3D space. This is illustrated in Fig. 3, with further demonstrations provided in the supplementary videos flag.mp4 and flag_triangle_soup.mp4. Figure 21 and the hand.mp4 video showcase examples of direct, manual interaction with the 3rd dimension. In Figure 20 (see 3D.mp4 in the supplementary materials), we demonstrate editing a shield that is partially occluded by a soldier. Our representation supports such transformation, which is why we describe it as intuitive to edit in 3D space.
> > >
> > > Additional visualizations where the camera angle changes can be found on:
> > > https://anonymous.4open.science/r/rebuttal-744F
> > >
> > > To improve visualization, we provide additional videos in the supplementary material: flag_simulation_with_trajectory.mp4 (previous flag.mp4) and flag_triangle_soup.mp4. We also include flag_simulation_with_camera_changing_and_trajectory.mp4, which demonstrate the effect of changing camera views. Through these visualization paths, we aim to clearly show how the camera movement enhances perception of the simulation. Additionally, we include a full-scene example with a lantern.mp4 to highlight the potential applications of our method.
> > >
> > > To illustrate practical applications, we reference the image of mountains or landscapes as an example of how a 2D object can be used. Notably, when using mountains as a background, it is not necessary to generate a full 3D model (e.g., via generative models), see train1.mp4 video in supplementary materials.
> > >
> > > Given that these clarifications enhance the reader’s understanding of the section on editability, we will integrate this discussion into the main paper.
> > >
> > > [1] FEYER, Stefan P., et al. 2D, 2.5 D, or 3D? an exploratory study on multilayer network visualisations in virtual reality. IEEE Transactions on Visualization and Computer Graphics, 2023, 30.1: 469-479.

---

### Official Review · Reviewer_VWK2 · 2025-03-12

**Overall Recommendation:** 3

**Summary:**

The paper introduces ​MiraGe, a method for representing and editing 2D images using parameterized 3D Gaussian components. By embedding 2D images in 3D space with flat Gaussians and leveraging mirror cameras for training, MiraGe achieves high-fidelity reconstruction and enables intuitive 3D-like editing (e.g., bending, physics-based animations). The method integrates with physics engines (e.g., Blender, Taichi) for realistic deformations and outperforms existing INR and Gaussian-based models (e.g., GaussianImage, SIREN) in reconstruction quality on Kodak and DIV2K datasets.

## Update after rebuttal
I thank the authors for their response. I have no further concerns and will maintain my positive score.

**Claims And Evidence:**

The authors claimed 1) state-of-the-art reconstruction quality, which is supported by the PSNR/MS-SSIM scores reported in Table 1; 2) 3D-like editing of 2D images, which is supported by Figures 3, 6, and 8 show manual edits (e.g., facial expressions, object bending) and physics-driven animations in Fig. 7.

**Essential References Not Discussed:**

The references are sufficient.

**Experimental Designs Or Analyses:**

Manual edits (Fig. 8) and physics animations (Fig. 7) are visually compelling but lack quantitative benchmarks. Many of the editing effects demonstrated in the paper and supplementary material appear to involve simple warping, which could also be achieved manually using traditional image editing tools like Photoshop. However, the proposed method offers a distinct advantage in its ability to integrate seamlessly with physics engines, enabling more dynamic and realistic modifications.

**Methods And Evaluation Criteria:**

​Methods:
The use of flat 3D Gaussians with GaMeS parametrization (from prior work) is innovative for bridging 2D/3D editing. Mirror cameras enhance spatial consistency, and three Gaussian control methods (Amorphous, 2D, Graphite) offer flexibility. However, physics integration is treated as a "black box," with no details on coupling Gaussians with engine dynamics.

​Evaluation:
Standard datasets (Kodak, DIV2K) and metrics (PSNR, MS-SSIM) are appropriate for reconstruction. However, editing capabilities are assessed only qualitatively. This paper only chose DragGAN and PhysGen as editing capabilities baselines and no non-generative baselines are compared in this paper.

**Other Comments Or Suggestions:**

No other comments.

**Other Strengths And Weaknesses:**

​Strengths:
1. Novel integration of 3D Gaussians for 2D image editing.
2. High reconstruction quality and support for intuitive 3D manipulations.
3. Demonstrated compatibility with physics engines.

​Weaknesses:
1. Limited quantitative evaluation of editing/physics realism.
2. Limited baseline methods for comparisons.
3. Most of the editing effects are relatively simple and subtle due to the method's non-generative nature, which limits its versatility and practical applicability.

**Questions For Authors:**

1. How does MiraGe handle ​complex edits (e.g., occlusions, texture changes) compared to traditional tools like Photoshop?
2. How does the model handle ​edge cases where significant modifications introduce visual inconsistencies?
3. How does MiraGe scale to ​high-resolution images in terms of training time and memory usage?

**Relation To Broader Scientific Literature:**

The proposed method combines the ​efficiency of Gaussian-based representations with editability, and integrates ​physics engines for realistic 2D image editing, expanding the application of physics-based simulations to traditionally 2D domains.

**Theoretical Claims:**

No theoretical proofs are provided for novel claims, but the method’s foundation in established techniques (3DGS, GaMeS) is reasonable.

---

> ### Author Rebuttal · Authors · 2025-03-29
>
> We appreciate the Reviewer's thoughtful feedback. We are pleased for the recognition of the distinct advantage of our proposed method in seamlessly integrating with physics engines, allowing for more dynamic and realistic modifications compared to traditional tools.
>
> W1/2
>
> In our paper, we address two tasks. The first, reconstruction, is evaluated using widely recognized benchmarks and metrics, as noted by Reviewer u88z, who found the methods and evaluation criteria appropriate.
>
> In the second editing task, we focused on qualitative assessment, acknowledging the limitations of our evaluation as the first to apply parametric 3D Gaussians to 2D images. Nevertheless, we compare our approach to generative methods to highlight differences in editing capabilities, aiming to introduce a novel 2D representation and inspire future advancements.
>
> W3
>
> In most generative methods, users have constrained control over edits. However, as shown in Fig. 10, our approach allows precise modifications, enabling users to adjust specific details. Additionally, we incorporate 3D representation and human perspective (Fig. 3), demonstrating that our method inherently possesses a 3D structure. This means that, beyond standard 2D transformations like rotation, we also have control over perspective.
>
> In PhysGen authors claim that “despite advancements in video generative models the incorporation of real-world physics principles into the video generation process remains largely unexplored and unsolved.” We show that our method can leverage physics engines, producing videos from 2D images without the need for generative model predicting the simulation (see visualization in supplementary materials). We show that using Taichi to apply materials such as sand to objects, as demonstrated in Fig.7.
>
> Q1
>
> Our goal is not to compete with tools like Photoshop but to highlight the advantages of representing 2D images with parameterized 3D Gaussians. While Photoshop allows for seamless image modifications, it relies on numerous hidden features such as specialized interpolation, imputation, and other corrections. When applying a simple affine transformation to an image, artifacts often emerge, necessitating pixel interpolation to correct them. In contrast, Gaussian components inherently preserve their structure under affine transformations, as an affine transformation of a Gaussian remains a Gaussian. For this reason, we believe directly comparing MiraGe to the Photoshop tool would be inappropriate.
>
> However similar to Photoshop layering, our image representation can incorporate the idea of image layers (as illustrated in Fig. 4). This allows for inpainting when parts of the image are occluded—for example, seamlessly inpainting in the background, as demonstrated in Fig. 11.
>
> It is worth mentioning that our model builds on features from 3DGS, enabling the application of existing Gaussian-based models. We believe that in practice, MiraGe can incorporate various style transfer methods for Gaussians. For instance, StyleGaussian [1c] allows for color adjustments during style transfer. However, such modifications fall outside the scope of this paper, as our focus is on shape editing rather than texture or color alterations.  We recognize this as a promising future research direction.
>
> [1c] K. Liu, et al. "StyleGaussian: Instant 3d style transfer with gaussian splatting." SIGGRAPH Asia 2024
>
> Q2
>
> Due to the character limitation, answer to this question is referenced in W1 and Q4, Reviewer u88z
>
> Q3
>
> We consider this an interesting question, as also noted by Reviewer u88z. We will address this in the camera-ready version.
>
> As part of our rebuttal, we conducted an experiment on a single image to illustrate how method scale. We selected a butterfly from the DIV2K dataset[2c]. For Training we used 100K initial Gaussians;5k, 10k, 30k iterations on the V100 GPU. In table we report compressed memory (MB) and training time (seconds).
>
> From the table below, we can observe that our method consistently achieves higher PSNR results compared to the baseline.
>
> [2c] https://data.vision.ee.ethz.ch/cvl/DIV2K/
>
> |Model||Our-5k iter|||Our-10k iter|||Our-30k iter|||GaussianImage|default settings|
> |---|---|---|---|---|---|---|:---|---|---|:---|---|---|
> ||PSNR|MB|Train time|⎮PSNR|MB|Train time|⎮PSNR|MB|Train time|⎮PSNR|MB|Train time|
> |1/none|30.76|3.56|209|⎮36.17|12.2|586|⎮47.04|35.65|2709|⎮28.34|2.41|189|
> |2|43.85|2.54|67|⎮47.21|4.20|158|⎮52.27|8.24| 651|⎮38.54|2.41|111|
> |3|51.51|2.41|45|⎮54.20|3.53|104|⎮58.81|5.66|406|⎮41.53|2.41|105|
> |4|56.86|2.36|40|⎮58.52|3.27|92|⎮65.02|4.83|341|⎮34.33|2.41|103|
>
> In practice, for high-resolution images, GS can be initially trained at a lower resolution and progressively refined to higher resolutions, following the Hierarchical GS strategy [3c], which we believe can be applied as a future work.
>
> [3c] B. Kerbl, et al. "A hierarchical 3d gaussian representation for real-time rendering of very large datasets." ACM Transactions on Graphics, 2024

---

### Official Review · Reviewer_u88z · 2025-03-14

**Overall Recommendation:** 3

**Summary:**

The paper proposes MiraGe, a novel approach for representing and editing 2D images using Gaussian Splatting. MiraGe uses flat-controlled Gaussian components positioned in 3D space, providing intuitive editing capabilities with a 3D perception. Key contributions include high-quality reconstruction results that outperform state-of-the-art methods such as GaussianImage, SIREN, and WIRE, and the integration with physics engines to enable physically plausible manipulations and realistic animations of 2D images. The method achieves notable improvements in reconstruction metrics (PSNR, MS-SSIM) compared to existing INR and 3D reconstruction approaches.

**Claims And Evidence:**

The claims of improved reconstruction quality, realistic 3D manipulation, and physics-based editing are generally well-supported by experiments on standard benchmarks (Kodak and DIV2K datasets).

**Essential References Not Discussed:**

The literature review is thorough, and essential references such as WIRE, NeuRBF, SIREN, GaussianImage, and I-NGP are well-discussed. However, 4D Gaussian Splatting (CVPR 2024), extending Gaussian Splatting to dynamic scenes, were not discussed and would further enrich the paper's context and comparison.

**Experimental Designs Or Analyses:**

The experimental design and analyses are sound. The evaluations include comparative studies against relevant baseline methods (GaussianImage, NeuRBF, I-NGP, WIRE, SIREN). However, deeper investigation into computational complexity, particularly the trade-offs in memory and computational efficiency compared to baseline methods, could strengthen the paper.

**Methods And Evaluation Criteria:**

The methods and evaluation criteria employed are appropriate and well-chosen. The paper uses widely recognized benchmarks (Kodak and DIV2K) and metrics (PSNR, MS-SSIM) suitable for evaluating the proposed method against competitors. Using these datasets facilitates direct comparison with previous work.

**Other Comments Or Suggestions:**

The paper is well-written and organized clearly.

**Other Strengths And Weaknesses:**

### Strengths:

- Clearly presents a novel and practical combination of Gaussian Splatting and 3D-like image editing.
- Demonstrates significant quantitative and qualitative improvements over state-of-the-art baselines.
- Integrates successfully with physics-based animation tools, extending applicability to various real-world use cases.

### Weaknesses:

- Significant edits produce visual artifacts, indicating potential limitations in practical usability without further refinement.
- The training time appears to be lengthy. While the paper includes a comparison between GaussianImage and 3DGS, incorporating comparisons with additional baselines would further highlight the strengths of this method.

**Questions For Authors:**

1. Could you discuss the computational overhead introduced by your approach compared to other baselines? Clarifying runtime performance or potential computational bottlenecks would influence the practical significance of the proposed method.
2. How does your method scale with image size or complexity (e.g., high-resolution images or more intricate scenes)? Providing additional insights or experiments on scalability would significantly improve the paper.
3. How well does your approach handle editing scenes with complex backgrounds? The examples in the supplementary materials appear somewhat simplistic—can you demonstrate editing effectiveness on images with more intricate backgrounds?
4. Can the artifacts arising from significant edits be mitigated systematically, or is the approach fundamentally limited by the Gaussian parametrization? A clear answer could help better understand the approach's limitations.

**Relation To Broader Scientific Literature:**

The paper positions itself well within existing literature by clearly explaining how it advances beyond GaussianImage's limitations of static representations. MiraGe leverages explicit Gaussian representations and integrates ideas inspired by recent developments in INRs and Gaussian Splatting's effectiveness in 3D representation. Thus, MiraGe contributes to both theoretical advancements in implicit neural representations and practical improvements in editable image representations.

**Theoretical Claims:**

The paper does not include explicit theoretical proofs or theoretical claims requiring validation. It mainly presents empirical evidence and algorithmic contributions.

---

> ### Author Rebuttal · Authors · 2025-03-29
>
> We thank the Reviewer for the feedback and constructive remarks regarding our paper that we believe will improve our paper. In particular, we are grateful for the Reviewer’s recognition that "the methods and evaluation criteria employed are appropriate and well-chosen".
>
> W1
>
> We acknowledge that significant edits can produce visual artifacts in the current version of our model and discuss it in in the Limitation Section. Nevertheless, our primary focus is on demonstrating the advantages of the proposed representation of 2D images using parametrized Gaussians and proving that such editing is feasible. We believe this work opens the door to numerous future research directions and improving even substantial edits.
>
> Q4
>
> It is a non-trivial question requiring substantial consideration.
>
> As the field of editing 3D Gaussians continues to develop, so do the methods for mitigating artifacts appearing in 3D objects. For example, a recent study (arXiv 03.2025) [1b] introduces a loss function that enforces Gaussians to maintain spherical shapes. In PhysGaussian the authors use Anisotropy Regularizer [2b]. In both cases this helps prevent the creation of "sharp" visible artifacts during rotations.
>
> In the context of 2D image “we introduced the mirror camera to ensure that Gaussians remain confined within a specific spatial region between the cameras, enhancing control and precision.” This approach effectively serves as a form of Gaussian regularization, as seen in Fig. 6 (first and second rows). To enhance readability, we will enlarge this figure.
>
> It is also important to highlight that artifacts may arise naturally when working with 2D representations, such as fading effects that occur with significant changes in perspective (e.g., rotations of 90 degrees in 3rd dimension). A useful analogy is a piece of paper or a photograph: when rotated 90 degrees in the third dimension, it effectively disappears from view.
>
> [1b] L Qiu, et al. ; “LHM: Large Animatable Human Reconstruction Model from a Single Image in Seconds”; arXiv  2025
>
> [2b] T. Xie, et al.; “PhysGaussian: Physics-Integrated 3D Gaussians for Generative Dynamics”;  CVPR 2024
>
>  W2
>
> Below we present the supplementation of Tab. 1 and a comparison with existing methods. The metrics for the baselines were taken from [3b]; however, we re-ran the experiments for GaussianImage (GI) -70K (GI-70K) and our using a V100 GPU to ensure a more reliable comparison. Those experiments are denoted in the following table by “*” next to the method name.
>
> |  | Kodak |  | Div2K |  |
> | --- | --- | --- | --- | --- |
> |  | PSNR | Train Time(s) | PSNR | Train Time(s) |
> | WIRE | 41.47 | 14339 | 35.64 | 25684 |
> | SIREN | 40.83 | 6582 | 39.08 | 15125 |
> | I-NGP | 43.88 | 491 | 37.06 | 676 |
> | NeuRBF | 43.78 | 992 | 38.60 | 1715 |
> | 3DGS | 43.69 | 340 | 39.36 | 481 |
> | GI-70K | 44.08 | 107 | 39.53 | 121 |
> | GI-70K* | 44.12 | 116 | 39.53 | 112 |
> | GI-100K* | 38.93 | 126 | 41.48 | 120 |
> | Our-70K; 5k iter | 49.07 |**57**| 44.37 |**75**|
> | Our-100K; 5k iter |  51.04 | 59 | 46.23 |79|
> | Our-70K; 30k iter | 57.41 | 547 | 53.22 | 789 |
> | Our-100K; 30k iter |**59.52**| 560 |**54.54**| 946 |
>
> We include the table with training time comparison including multiple baseline methods. The time provided for our method is the full training time (30000  and 5000 iteration steps).  Additionally, using the butterfly image from the DIV2K dataset as an example, we demonstrated that our method (Our-100K) achieves higher PSNR than GI in just 30 seconds, see Fig. 3.  This is also supported by our method surpassing GI in PSNR with only 5k iterations, while also having a shorter training time.
>
> Q1
>
> While the inference of our method is fast, the storage cost introduced by the original 3DGS representation used by our method is high. To overcome this we  use the existing compression tool spz [3b]. This allows us to reduce the memory overhead by up to 95%; please see Tab. 4 in supplementary material. Additionally we include GI as our baseline. Note that GI compresses the Gaussian representation as part of their pipeline, and spz compression algorithm is not applicable to their representation:
>
> |  | Camera setting | PSNR | Memory(MB) | Compressed memory (MB) |
> | --- | --- | --- | --- | --- |
> | GI (70k) * | - | 44.12 | - | 2.41 |
> | GI (100k)* | - | 38.93 | -| 3.44 |
> | Our (100k) | A/ One camera | 51.56 | 31.25 | 2.42 |
> | Our (100k) | A/ Mirror camera | 59.52 | 117.25 | 7.80 |
>
> [3b] https://github.com/nianticlabs/spz
>
> Q2
>
> Due to the character limitation, answer to this question  is referenced in Q3, Reviewer VWK2
>
> Q3
>
> Indeed, in complex scenes with intricate backgrounds, the task becomes more challenging, as we mention in the Limitations Section. However, as shown in Fig. 11, our method allows for seamless object manipulation, such as repositioning a flower using a physic engine such as Taichi Elements, while effectively applying inpainting to ensure a gap-free result.

---

### Official Review · Reviewer_bkJM · 2025-03-14

**Overall Recommendation:** 4

**Summary:**

The paper introduces MiraGe, a novel method for representing and editing 2D images using flat 3D Gaussian components. The approach leverages Gaussian splatting in 3D space to enable high-quality image reconstruction and realistic editing capabilities. MiraGe allows for both 2D and 3D manipulations of images, creating the illusion of 3D transformations while maintaining the integrity of the original 2D image. The method employs parameterized flat Gaussians and integrates with physics engines for dynamic modifications. Experimental results demonstrate state-of-the-art performance in image reconstruction quality and editing capabilities.

## Update after rebuttal

Thanks for the author's rebuttal. It has well addressed my concerns.  I suggest accept this work.

**Claims And Evidence:**

The paper makes several claims about the effectiveness of **MiraGe** in image reconstruction and editing, which are well-supported by the following evidence:

- Quantitative results showing improvements in PSNR and MS-SSIM metrics compared to previous methods (Table 1).
- Visual comparisons demonstrating better reconstruction quality and fewer artifacts than competing approaches (Figures 5, 8, 10).
- Ablation studies highlighting the impact of different model components, such as the mirror camera setup (Table 2).
- Demonstrations of editing capabilities, including manual modifications and physics-based animations (Figures 3, 7, 18, 19).

The evidence convincingly supports the claims made. The experiments cover multiple datasets and provide both quantitative and qualitative comparisons. The ablation studies effectively isolate the contributions of different components of the proposed framework.

**Essential References Not Discussed:**

N/A

**Experimental Designs Or Analyses:**

The experimental setup is comprehensive.

And also, the authors compare their method with the generative approach Drag-GAN. Despite Drag-GAN being a generative model with extensive prior knowledge, this work still achieves comparable performance, which is very promising.

**Methods And Evaluation Criteria:**

The method is clear, and the evaluation criteria are well-defined. It includes many quantitative metrics, along with qualitative comparisons to other methods, making the experiments very thorough.

**Other Comments Or Suggestions:**

Does the MiraGe structure require training on a single image each time? Would this be considered low-efficiency? What plans do the authors have for improving this aspect in the future?

**Other Strengths And Weaknesses:**

A small problem is that the author could add the PSNR/SSIM score on Fig.5, it would be more clear to see the advantage.

Overall I believe this is a great work.

**Questions For Authors:**

If more physical factors, such as real-world lighting, BRDF, and materials, are considered, would this enhance the performance of the work? If it possible to incorporate these factors to improve MiraGe's performance? In other words, does MiraGe have the potential to expand in the direction of inverse rendering (i.e. NeRFactor, Ref-NeRF)?

**Relation To Broader Scientific Literature:**

This work demonstrates many applications. Compared to previous work, it offers better image representation and manipulation capabilities than Gaussian-Image (ECCV 2024). Unlike Drag-GAN (SIGGRAPH 2023), it doesn't require as much prior knowledge.

Additionally, it can integrate two images (Fig. 4), suggesting that it may also have potential for image harmonization, though real-world lighting conditions might add complexity.

**Theoretical Claims:**

The paper does not present extensive theoretical proofs but rather focuses on conceptual framework and experimental validation.

---

> ### Author Rebuttal · Authors · 2025-03-29
>
> We sincerely thank the Reviewer for their valuable feedback and are pleased with the appreciation of our work. In particular, we are especially grateful for the recognition of the breadth and depth of our experiments "covering multiple datasets and providing both quantitative and qualitative comparisons."
>
> W1. A small problem is that the author could add the PSNR/SSIM score on Fig.5.
>
> Thank you for this suggestion. We will add the mentioned metrics to the Fig. 5 in the camera-ready version:
>
> | name | PSNR | SSIM |
> | --- | --- | --- |
> | GaussianImage | 43.73 | 0.9993 |
> | MiraGe | 62.00 | 0.9999 |
>
> C&S1. Does the MiraGe structure require training on a single image each time? Would this be considered low-efficiency?
>
> The pipeline does require training for each image individually. However, it is important to note that even with just 30 seconds of training, our method already produces better results compared to existing approaches (Fig. 9; Our-100k, using the butterfly image from DIV2K as an example). At the same time, longer training is beneficial for more-optimal PSNR performance. For more comparison, please refer to our answer to Reviewer u88z, W2 where we show that our method is surpassing baselines in PSNR with only 5k iterations, while also having a shorter training time.
>
> C&S2. What plans do the authors have for improving this aspect in the future?
>
> A possible direction to improve our approach with regards to training a different model for each individual image, could be the use of a generative model to prepare the collection of Gaussians based on the input image. While literature explores such an approach to the best of our knowledge current models work on foreground objects, and not full scenes like [1a]. In our case the images used are real life photographs, where both foreground and background are required to be reconstructed. Additionally, we would like to highlight that our model rapidly obtains high quality reconstruction.
>
> [1a] TANG, Jiaxiang, et al. DreamGaussian: Generative Gaussian Splatting for Efficient 3D Content Creation. In: The Twelfth International Conference on Learning Representations.
>
> Q1. If more physical factors, such as real-world lighting, BRDF, and materials, are considered, would this enhance the performance of the work? If it possible to incorporate these factors to improve MiraGe's performance? In other words, does MiraGe have the potential to expand in the direction of inverse rendering (i.e. NeRFactor, Ref-NeRF)?
>
> Thank you for posting this interesting question. In literature there exist works that incorporate real-world lighting using 3D Gaussian Splatting [2a, 3a]. In contest to BRDF, 3D Gaussian Ray Tracing [4a] uses Ray Tracing to enable secondary lightning effects. Since all the mentioned methods (including ours) are based on 3D Gaussian Splatting, we believe that integrating similar ideas into MiraGe is possible. We consider this a promising direction for future work, which we will post in the main manuscript.
>
> [2a] J. Gao, et al. “Relightable 3D Gaussians: Realistic Point Cloud Relighting with BRDF Decomposition and Ray Tracing”; ECCV2024
>
> [3a] Z. Bi, et al. GS^3: “Efficient Relighting with Triple Gaussian Splatting”; SIGGRAPH Asia 2024
>
> [4a] N. Moenne-Loccoz, et al.; “3D Gaussian Ray Tracing: Fast Tracing of Particle Scenes”; SIGGRAPH Asia 2024

---

### Decision · Program_Chairs · 2025-05-01

**Decision:**

Accept (poster)

**Comment:**

There were three acceptance and one rejection recommendations.
After the rebuttal, the rejecting reviewer maintained their recommendation but also agreed to accept the paper for publication.
After careful consideration of the reviews, the paper was accepted. However, the authors are encouraged to address the limitations or trade-offs pointed out by the reviewer.